

# Hydrodynamics of the interacting Bose gas
# in the Quantum Newton Cradle setup

Jean-Sébastien Caux [1\*], Benjamin Doyon [2], Jérôme Dubail [3],
Robert Konik [4] and Takato Yoshimura [5]

**1** Institute for Theoretical Physics Amsterdam and Delta Institute for Theoretical Physics, University of Amsterdam, Science Park 904, 1098 XH Amsterdam, The Netherlands
**2** Department of Mathematics, King's College London, Strand, London WC2R 2LS, UK
**3** CNRS & IJL-UMR 7198, Université de Lorraine, F-54506 Vandoeuvre-lès-Nancy, France
**4** Condensed Matter and Materials Science Division, Brookhaven National Laboratory, Upton, NY 11973 USA
**5** Department of Mathematics, King's College London, Strand, London WC2R 2LS, UK

⋆ J.S.Caux@uva.nl

## Abstract

**Describing and understanding the motion of quantum gases out of equilibrium is one of the most important modern challenges for theorists. In the groundbreaking Quantum Newton Cradle experiment [Kinoshita, Wenger and Weiss, Nature 440, 900 (2006)], quasi-one-dimensional cold atom gases were observed with unprecedented accuracy, providing impetus for many developments on the effects of low dimensionality in out-of-equilibrium physics. But it is only recently that the theory of generalized hydrodynamics has provided the adequate tools for a numerically efficient description. Using it, we give a complete numerical study of the time evolution of an ultracold atomic gas in this setup, in an interacting parameter regime close to that of the original experiment. We evaluate the full evolving phase-space distribution of particles. We simulate oscillations due to the harmonic trap, the collision of clouds without thermalization, and observe a small elongation of the actual oscillation period and cloud deformations due to many-body dephasing. We also analyze the effects of weak anharmonicity. In the experiment, measurements are made after release from the one-dimensional trap. We evaluate the gas density curves after such a release, characterizing the actual time necessary for reaching the asymptotic state where the integrable quasi-particle momentum distribution function emerges.**

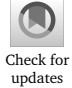

## Contents



# 1   Introduction

In 2006, the pioneering experiment of the "Quantum Newton Cradle" [1] (QNC) provided a groundbreaking demonstration of the fundamental importance of the large number of conservation laws in the description of out-of-equilibrium one-dimensional (1d) quantum systems. In this experiment, an ultracold gas of Rubidium atoms is confined to one dimension by a strong transverse optical trap, and weakly confined to a finite region by a longitudinal quasi-harmonic potential. A sequence of Bragg pulses imparts a linear combination of oppositely-directed momenta to the initial cloud that lies at the center of the trap. After a short dephasing period, two independent clouds emerge, which oscillate within the trap and meet twice every period.

Surprisingly, upon meeting and interacting, the clouds do not thermalize to a single zero-momentum cloud, as would happen in an ordinary gas. Instead, the two clouds re-emerge and continue their oscillations. This is likened to the Newton cradle, the famous desktop toy in which, upon collision, momentum is transferred between the end beads.

To a good approximation [2], the dynamics of a 1d gas of $N$ identical bosonic atoms with mass $m$ at positions $x_i$ is described by a hamiltonian with "delta" repulsion called the Lieb-Liniger model [3,4],

$$H = -\frac{\hbar^2}{2m} \sum_{i=1}^{N} \partial_{x_i}^2 + g \sum_{i<j} \delta(x_i - x_j) + \sum_{i=1}^{N} V(x_i), \tag{1}$$

where $g > 0$ is the repulsion strength and $V(x)$ the longitudinal trapping potential. In the absence of a potential $V(x)$, this model is integrable —it has an extensive number of conserved quantities—, a property that was conjectured to be at the root of the lack of thermalization in the QNC experiment. This has given rise to a wealth of theoretical developments on the generalization of thermalization in integrable models, following Refs. [5,6]. It was understood that, even after very long times, integrable systems fail to converge to a thermal Gibbs state. Instead, they reach a macrostate that is entirely characterized by the distribution of quasi-particles [7], in a way that parallels and generalizes the early work of Yang and Yang on the thermodynamics of the 1d Bose gas [8].

Realistic theoretical modeling of the QNC experiment requires to deal with $N \sim 10^2$-$10^3$ particles at finite repulsion strength —i.e. away from the hard-core limit $g \to +\infty$ that can be treated with exact determinantal methods [9]—and with an inhomogeneous potential $V(x)$. This has remained completely out of reach of modern theoretical tools, and has represented one of the most prominent challenges in quantum many-body theory in the past decade. On the analytical side, the difficulty is twofold. First, it is known that the external potential breaks integrability. How, then, is the physics of integrability coming into play? Second, the setup is highly inhomogeneous, which is a major issue for most analytical techniques available in 1d [10]. On the numerical side, the situation is not better: modern tools for out-of-equilibrium quantum many-body physics like tDMRG [11] are limited to small numbers of particles and small times in QNC-like setups [12], while numerical methods based on integrability [13–17] break down because of the strong inhomogeneity.

A new set of theoretical tools and ideas have come to the fore in 2016 that, as we argue here, provide such a realistic modeling. These pertain to the theory of "generalized hydrodynamics" (GHD) [18,19], a hydrodynamic approach to 1d integrable models that is able to account for a wide variety of inhomogeneous situations [20], including states obtained from domain-wall initial boundary conditions [18,19,21–23], the effects of external potentials [24] and the propagation of waves [25,26]. The last two of these works show excellent agreement with full numerical simulations of the Lieb-Liniger model for $N \sim 40$ particles [25] based on NRG+ABACUS [13–17,27], and with tDMRG in the XXZ chain [26]. The goal of this Letter is to apply the newly developed GHD framework to provide a *quantitatively* accurate modeling of the QNC experiment that remains easily tractable numerically.

## 2   The GHD equation

GHD is a hydrodynamic approach that captures the behavior of the Lieb-Liniger model —as well as any other Bethe ansatz integrable model [18,19]— at the Euler scale [28]. This is the scale at which variations of densities of particles, momentum, energy, and all other local conserved quantities, are slow enough, in both space and time. The system is then viewed as an assembly of "mescoscopic" fluid cells at spacetime positions $(x, t)$, each of which is

large enough such that it contains a thermodynamically large number of bosons, and small enough such that the gas is homogeneous throughout the cell. As in any other hydrodynamic theory [28], at this scale GHD assumes that local maximization of entropy has occurred. In standard approaches [12, 29–32], this would imply that each fluid cell is locally in a Galilean boost of a thermal Gibbs state. Instead, GHD keeps track of the infinite set of conserved charges by representing the local macrostate by its distribution $\rho_p(\theta)$ of quasi-particles with velocity $\theta$. In GHD, this distribution is position- and time-dependent, and is denoted $\rho_p(\theta, x, t)$. In terms of the quasi-particles, the density of bosons $n(x, t) = \left\langle e^{iHt} \left( \sum_{j=1}^N \delta(x - x_j) \right) e^{-iHt} \right\rangle$ is recovered, in the thermodynamic limit by integrating locally over all the quasi-particles:

$$n(x, t) = \int d\theta \rho_p(\theta, x, t). \tag{2}$$

Similarly, all other densities of conserved charges in the Lieb-Liniger model, such as momentum, energy, or others, may be expressed as integrals $\int d\theta h(\theta) \rho_p(\theta, x, t)$ for suitably chosen functions $h(\theta)$, see e.g. Ref. [18].

The conservation of quasi-particle densities exchanged between neighbouring fluid cells fully fixes the dynamics [18, 19]. Taking into account the trapping potential $V(x)$, the GHD equation reads [24]

$$\partial_t \rho_p + \partial_x [v^{\text{eff}} \rho_p] = \left( \frac{\partial_x V}{m} \right) \partial_\theta \rho_p, \tag{3}$$

where the effective velocity $v^{\text{eff}}$ itself depends on the local distribution of quasi-particles $\rho_p(., x, t)$ through the "dressed" functions,

$$v^{\text{eff}}(\theta) = \text{id}^{\text{dr}}(\theta)/1^{\text{dr}}(\theta). \tag{4}$$

Here $\text{id}(\theta) = \theta$ and $1(\theta) = 1$, and the "dressing" is defined for any function $f(\theta)$ as the solution of the linear integral equation $f^{\text{dr}}(\theta) = f(\theta) + \int d\theta' \frac{2g/m}{(g/\hbar)^2 + (\theta - \theta')^2} \frac{\rho_p(\theta')}{1^{\text{dr}}(\theta')} f^{\text{dr}}(\theta')$. Thus, GHD is a large-scale approach to the (inhomogeneous) Lieb-Liniger model (1) that requires to (i) specify an initial condition $\rho_p(\theta, x, 0)$ at $t = 0$, and (ii) solve the partial differential equation (3).

A number of techniques are available to numerically solve Eq. (3). Numerical methods for solving partial differential equations [33] can be adapted to GHD —a discussion about this can be found in Ref. [26]—. Other methods have been discovered in the past months, that can be easier to implement, and in certain simple cases, possibly more efficient (an example is an exact solution expressed as a system of integral equations [34], which can be solved recursively on a computer very quickly; but until now this applies only to the case without an external potential $V(x)$).

An efficient technique [25] is that of the zero-entropy subspace. It uses the simplification of GHD for initial states which have zero Yang-Yang entropy, such as zero-temperature states. In this case, the solution is expressed via space-time dependent Fermi points, whose equations are simple enough to be directly amenable to numerical solution. Zero-entropy GHD works in the presence of external potentials, and it also sheds light on phenomena such as shock formation and dissolution [25]. Zero-temperature initial states are often a good first approximation for low-temperature experimental setups. It is possible to analyze such a zero-temperature version of the QNC by taking the initial state as the ground state in a double-well potential that splits the gas into two. By releasing the two clouds in a single harmonic well, the main phenomenon – the lack of thermalization upon cloud collisions – of the QNC experiment is observed (see Appendix I).

Finally, the technique which we chose to use here is a classical molecular simulator [35]. This is essentially a Monte Carlo technique: a gas of classical particles [36] is initially sampled

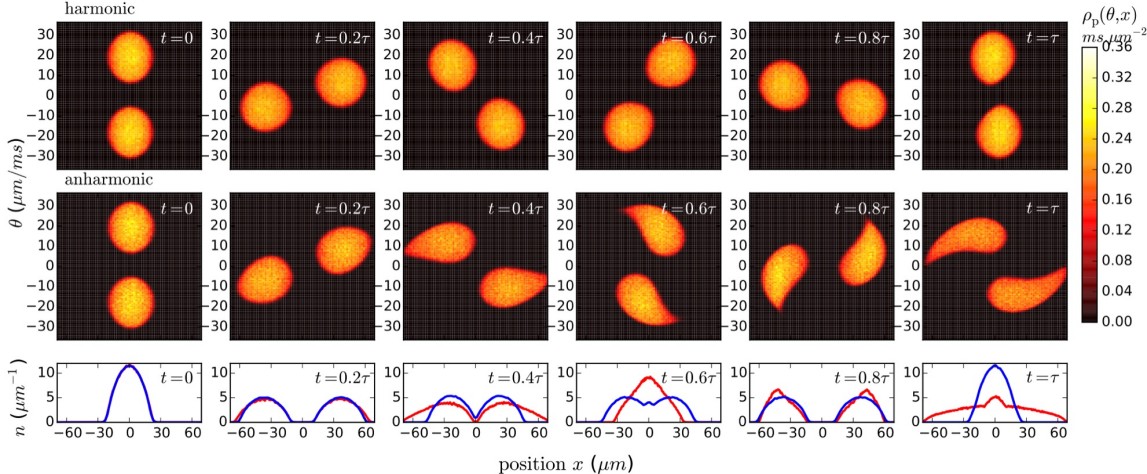

Figure 1: Evolution of the density of quasi-particles $\rho_{\mathrm{p}}(\theta, x, t)$ —here plotted in the $(x, \theta)$-plane— in the QNC setup, with parameters given in the text. The solution of the GHD equations are obtained from the flea gas [35]. The results are displayed for the harmonic trap (top row) and the weakly anharmonic one (middle row), on one period of the (quasi-)harmonic trap. (Bottom row) Corresponding density of particles $n(x, t)$, for the harmonic trap (blue) and the anharmonic one (red).

according to the distribution of integrable quasi-particles determined, via appropriate integral equations, by the initial quantum state, and then let to evolve in a deterministic fashion by a specific dynamics that encodes the interaction of the quantum gas. The hydrodynamic description of this classical gas is, at the Euler scale, exactly the same GHD equation (3) as the one found in the quantum gas [35]. This quantum-classical equivalence provides an extremely efficient method for simulating solutions to the GHD equation, that is able to account simultaneously for arbitrary initial conditions and for external potentials.

## 3 Modeling the Bragg pulse sequence

We start from the thermal Gibbs state in a (quasi-)harmonic potential $V(x)$. The exact experimental distribution pre-pulse is not known, and is not expected to be exactly thermal as cooling methods deplete the large-momentum region. However a thermal distribution is expected to be a good approximation, accounting well enough for the remaining energy that brings the system away from absolute zero temperature. The pre-pulse density $\rho_{\mathrm{p}}^{\mathrm{Gibbs}}(\theta, x)$ is obtained by searching for the finite-temperature hydrostatic solution of (3), which can be shown [24] to be equivalent to a local density approximation (LDA) [37], obtained using the Thermodynamic Bethe Ansatz [8]. This technique is known to give an accurate description of atomic gases at equilibrium in quasi-harmonic traps [38, 39]. We model the post-pulse distribution $\rho_{\mathrm{p}}(\theta, x, 0)$ by imparting, in a random fashion, positive and negative momenta to the quasi-particles. More precisely,

$$\rho_{\mathrm{p}}(\theta, x, 0) = \frac{1}{2}\rho_{\mathrm{p}}^{\mathrm{Gibbs}}(\theta + \theta_{\mathrm{Bragg}}, x) + \frac{1}{2}\rho_{\mathrm{p}}^{\mathrm{Gibbs}}(\theta - \theta_{\mathrm{Bragg}}, x), \tag{5}$$

corresponding to a pulse where the momentum of a quasi-particle is kicked by $\pm m\theta_{\mathrm{Bragg}}$ with equal probability $\frac{1}{2}$. In fact, results of [40] show that the momentum distribution function (MDF) of *particles* (both the real bosonic particles, and their fermionic Jordan-Wigner transform) are affected in this way by a Bragg pulse. To a good approximation the same holds

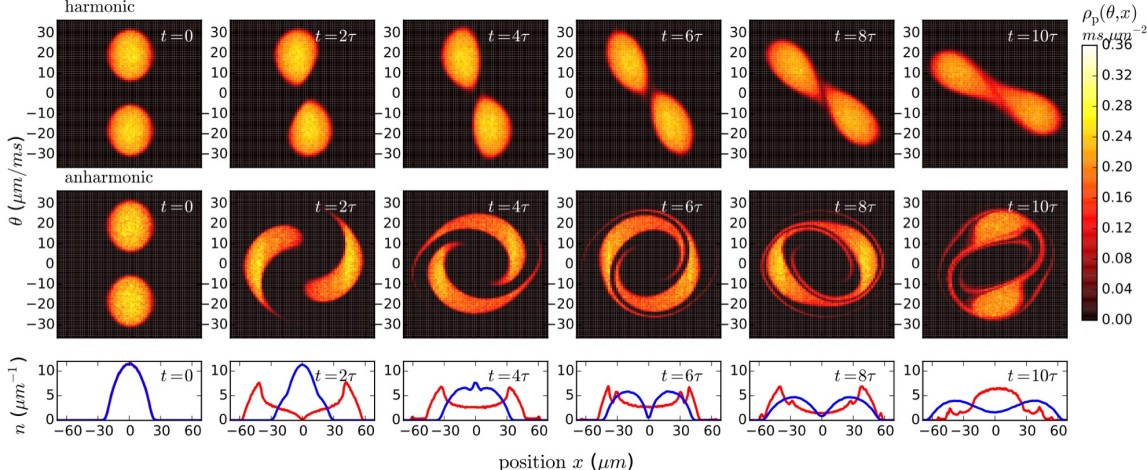

Figure 2: Same as Fig. 1, on a larger time window. In the harmonic case, the two blobs in the $(x, \theta)$-plane keep rotating around each other after several trap periods. In the anharmonic case, the distribution $\rho_p(\theta, x)$ is strongly stirred up after a few trap periods, and it goes to stationary state that looks rotationally invariant in the $(x, \theta)$-plane.

for the quasi-particle MDF because of the large momentum difference between clouds: the inter-cloud interaction is therefore effectively hard-core, hence screened for the fermions (the intra-cloud uniform momentum kicks by $\pm\theta_{\text{Bragg}}$ are Galilean transformations). We have also considered more refined models for the Bragg pulse (see Ref. [40] for a detailed discussion), but we find that it does not affect drastically the evolution at later times, and this simple version is already in good agreement with what is seen in the experiment.

## 4    Results for the (harmonic) QNC

We work with the following parameters, which are close to the ones given in Ref. [1]. We take $N = 350$ bosonic atoms with mass $m = 142.9 \times 10^{-27}$kg in a harmonic trap $V(x) = m\omega^2 x^2/2$ with period $\tau = \frac{2\pi}{\omega} = 13$ms, and with repulsion strength $c = \frac{mg}{\hbar^2} = 12.3\mu\text{m}^{-1}$. The pre-pulse state is a thermal Gibbs state at temperature $T = 10\mu\text{m}^{-2} \times \hbar^2/m \simeq 57$nK. We then apply the Bragg pulse sequence (random kick of the particle velocities by $\pm\theta_{\text{Bragg}} = \pm\hbar k_{\text{Bragg}}/m$) with $2k_{\text{Bragg}} = 25\mu\text{m}^{-1}$. In the initial state, the density at the center of the trap is $n(0) = 11.5\mu\text{m}^{-1}$, which gives a dimensionless interaction strength $\gamma = c/n(0) \simeq 1.07$, close to the experimental value [1]. Notice that we are far from both the hard-core (or Tonks-Girardeau) limit $\gamma \gg 1$ and the weakly interacting (or Gross-Pitaevskii) limit $\gamma \ll 1$.

After the Bragg pulse, the dynamics of the Bose gas is given by Eq. (3), which we solve with the molecular dynamics simulation, see Fig. 1. We observe that the two blobs, originally separated in momentum space and symmetric with respect to $\theta \mapsto -\theta$, evolve by performing a deformed rotation-like movement around the origin of phase space. At time $t \simeq \tau/4$, the two blobs have zero spatial overlap, corresponding to two well separated clouds in real space. At time $t \simeq \tau/2$, they have again overlapping spatial support. Their evolution is not drastically affected by this overlapping, and it is clear, in the phase space picture, how the two atomic clouds can pass through each other. The actual gas density —bottom row of Fig. 1— is obtained by integrating $\rho_p(\theta, x, t)$ over $\theta$ according to Eq. (2).

We observe that the blobs are slightly slowed down when they overlap: it takes them

slightly longer than a period $\tau$ to come back to their original position along the vertical axis. The inter-cloud interaction at spatial overlap is weak because of large momentum separation, but nonzero, with perceptible effect. Further, after several periods, the blobs elongate transversally, roughly towards the center of rotation in phase space, see Fig. 2. This slow "many-body dephasing" effect, controlled by how far the effective velocity is from the bare velocity [1], is also due to inter-cloud interactions. Notice how, in Fig. 1, the blobs' shape stay relatively unchanged until just before half a period, $t = 0.4\tau$, while after the clouds have passed through each other, $t = 0.6\tau$, slight modifications have occurred. Particles of one cloud, going through the other cloud, are scattered on the scale of the scattering length. We observe that the particle density spreads mostly towards lower energies. The spreading must indeed be stronger at lower energies than at higher energies, because total energy is conserved, while the change in energy per distance (per momentum) is greater at higher energies than it is at lower energies (as is clear from the form of the potential, and of the kinetic energy as function of momentum). Intra-cloud scattering is also present but its effect is weaker as fewer scattering events occur.

## 5 Effects of weak anharmonicity

In the QNC experiment, the trapping potential is not exactly harmonic. To study the effects of anharmonicity, we now replace the harmonic trap by $V(x) = \frac{1}{\pi^2}\omega^2\ell^2(1 - \cos(\frac{\pi x}{\ell}))$, both before and after the Bragg pulse. This form is chosen in order to mimic actual experimental setups, where potentials are often close to trigonometric functions. In particular, the anharmonicity has the property that the potential is smaller than harmonic further away from the centre (i.e. it becomes flatter). We chose $\ell = 100\mu m$, and all other parameters are identical to the ones of the harmonic case. In Fig. 1 (second row), we see that the two blobs get deformed much more quickly than in the harmonic case; the distribution $\rho_p(\theta, x, t)$ gets more and more stirred up after few periods (Fig. 2, second row). A similar effect was recently observed for a single particle in an anharmonic trap in Ref. [41]. In particular, we observe in Figs. 1 and 2 (second row) that the blobs elongate longitudinally, roughly in the direction of their motion in phase space. This is an effect of the anharmonicity $\ell < \infty$: because the potential becomes flatter far from the origin, particles with higher velocities take longer to come back to their original position. This "single-body dephasing" effect is well captured by the spreading of blobs of *noninteracting* particles, and can be quantified by evaluating the nonzero difference $\Delta t$ between the periods of independent particles of different velocities. With minimal (maximal) velocity of $10\mu m/ms$ ($40\mu m/ms$) within one initial blob, one finds $(\Delta t)_{max} \approx 1.2ms$. The anharmonic mixing time is approximately $\tau \cdot \tau/(\Delta t)_{max} \approx 11\tau$, in agreement with Fig. 2. Many-body dephasing is also present in the anharmonic case. Noticeably, without interactions the original blobs would simply disintegrate into long spiraling filaments. We see instead at $10\tau$ new structures forming. These appear as many-body dephasing takes effect, causing filaments to merge and high-energy (longer-period) tails to scatter to lower energies, opening the way for the quicker single-body effects to reform new blobs.

## 6 Discussion: thermalization?

We now turn to the fundamental question that was raised in 2006 in Ref. [1]: does the gas thermalize after a sufficiently large number of oscillations? We first consider this question

---

[1]See Appendix II for expressions of the time variation of the total number of quasi-particles, and of their total energy, within the region $m\theta^2/2 + V(x) < E$, showing non-conservation in the presence of interaction, controlled by $|v^{\text{eff}}(\theta) - \theta|$ for $\theta$ without the region.

within pure GHD. Within this context, the answer is negative. The reason is that, in addition to its particle number and its global energy

$$E = \int dx\, d\theta \left(\frac{m\theta^2}{2} + V(x)\right)\rho_{\mathrm{p}}(\theta, x), \qquad (6)$$

GHD keeps track of many initial features. Indeed, we found that even in the presence of a trapping potential $V(x)$, the GHD equation (3) possesses infinitely many conserved quantities $Q(\eta)$, with a continuous parameter $\eta \in [0,1]$ – see Appendix III, that give rise to conservation of the Yang-Yang entropy [8] and generalizations thereof.

This is incompatible with the system converging to a Gibbs state, even at infinite time. Much like phase-space preservation in classical mechanics may lead to fractal trajectories, in GHD these constraints give rise to fine structures developing at ever decreasing scales; see for instance the times of order $t \simeq 6\tau - 10\tau$ in the anharmonic case of Fig. 2, Ref. [41] for a similar study in the hard-core ($\gamma \to \infty$) limit, and the very recent work on the dynamics of a classical hard rod gas in a trap [42]. However, there is a sense in which GHD converges to a smooth, stationary state: this is through coarse-graining.

## 7 Microscopics and coarse-graining

GHD is an idealized hydrodynamic description with no UV cutoff. In contrast, the 1d Bose gas (1) is a microscopic model described by GHD only at larger scales, and is therefore only an imperfect realization of GHD.

Now GHD predicts the appearance of fine structure in phase space, i.e. strong variations of $\rho_{\mathrm{p}}(\theta, x)$. The way the microscopic model treats the appearance of these UV degrees of freedom in GHD amounts to coarse graining: the density $\rho_{\mathrm{p}}(\theta, x)$ predicted from GHD gets essentially replaced by its average over fluid cells $[x, x+dx]$ of size larger than the inter-particle distance, so eliminating the UV degrees of freedom from the problem. As a result of this coarse-graining, the entropy of the gas increases and the quantities $Q(\eta)$ are not conserved. Interestingly, this effect was studied in Ref. [42] for the classical hard-rod gas [28], and was marked as the consequence of an interim chaotic regime.

In an attempt to understand better coarse-graining effects, we have discovered that under certain hydrodynamic conditions, the GHD equations are *invariant under the coarse-graining procedure* where quasi-particle phase-space densities are replaced by their local averages – see Appendix IV. In particular, GHD time evolution commutes with this coarse-graining procedure, as long as the coarse-gaining cells are mesoscopic and certain dephasing conditions hold. This suggests an amount of universality to the GHD solutions; for instance, as fine structures emerge, one may coarse-grain, still correctly describing the solution over time.

We have not been able to study coarse-graining effects directly in the 1d Bose gas (1), as it would require a full quantum simulation which is currently beyond reach. However, the loss of fine structures at large times is made clear as we vary the UV cutoff in our molecular simulation (Appendix IV). The UV cutoff can be taken as the number of classical particles used in the simulation, and this provides an indication for the relation between loss of fine structures and actual particle numbers in the 1d Bose gas. We also observed that at a coarse-grain level, a stationary state emerges – see Appendix V.

Although not a Gibbs state, as argued in [24] such a stationary state must be a "Generalized HydroStatics" solution: it satisfies $\partial_x[v^{\mathrm{eff}}\rho_{\mathrm{p}}] = (\partial_x V/m)\partial_\theta\rho_{\mathrm{p}}$ (this was explicitly checked in the hard-rod case in [42]). We furthermore have partial numerical evidence that such a stationary state is a universal property of coarse-grained GHD, and not sensitive to particular microscopic realizations of GHD.

# 8 Profiles after Time-Of-Flight (TOF) in 1d

In Ref. [1], it is not the *in situ* density that is being measured, but the density profile after a trap release in 1d: in order to have a cloud that is large enough compared to the resolution of the camera, the longitudinal potential is suddenly released, and the atomic cloud expands for a time $t_{\text{TOF}}$, before a picture is taken. When $t_{\text{TOF}}$ is sufficiently large, the real-space density profile $n_{\text{TOF}}(x)$ is directly related to the momentum distribution function (MDF) of integrable quasi-particles $\rho_{\text{p}}(\theta, x')$ *just before the expansion*, as

$$n_{\text{TOF}}(x) \underset{t_{\text{TOF}} \to \infty}{=} \frac{1}{t_{\text{TOF}}} \int dx' \rho_{\text{p}}(\theta, x'), \qquad \theta = x/t_{\text{TOF}} \tag{7}$$

(see Refs. [43–48] and Appendix VI). However, importantly, it is not only the asymptotic distribution for $t_{\text{TOF}} \to \infty$ that is accessible thanks to GHD. Instead, the full expansion of the cloud can be simulated by suddenly setting $V = 0$ in Eq. (3) at the expansion time. Since, during the whole expansion, the typical length scale of density variations grows proportionally to the typical inter-particle distance, the hydrodynamic approximation remains valid throughout. Hence, GHD is particularly well-suited for predicting the outcome of such measurements, even at finite Time-Of-Flights. To our knowledge, this is not accessible by other techniques.

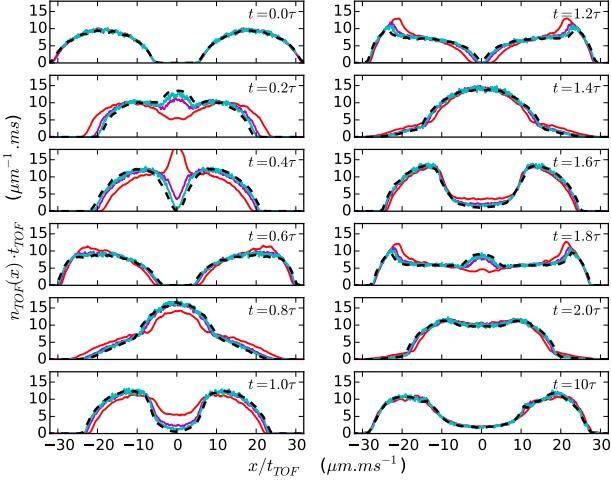

Figure 3: Density profile after Time-Of-Flight $t_{\text{TOF}}$ in 1$d$, for the QNC in an anharmonic trap with the same parameters as in Figs. 1-2. We compare the profiles for $t_{\text{TOF}} = 10$ms (red), $t_{\text{TOF}} = 25$ms (magenta), $t_{\text{TOF}} = 50$ms (cyan) and $t_{\text{TOF}} = \infty$ (dashed black, corresponding to Eq. (7)).

We have evaluated the expansion curves obtained after 10$ms$, 25$ms$ and 50$ms$, as released from an evolution within the anharmonic potential for times from 0 to 2$\tau$, and at time 10$\tau$, see Fig. 3. The time necessary to reach the asymptotic state (7) depends strongly on the initial distribution. It is very short for distributions with high momenta such as just after the Bragg pulses (less than 5$ms$ for an expansion at $t = 0\tau$), but it is much longer when particles are slower and mostly lie at the edges of the potential (requiring more than 60 ms at $t = 0.25\tau$). This corresponds to an expanded cloud that is 10 (for initially fast particles) to 30 (for initially slow particles) times larger than the original one.

## 9 3d expansion and momentum distribution

Longitudinal expansions are in strong contrast with three-dimensional trap releases. In the latter case, the profile after a long TOF is given by the MDF of the real physical bosons, as opposed to that of quasi-particles. It is also possible to calculate the bosonic MDF, but this is more difficult. An approximation scheme combining GHD data with the ABACUS algorithm is proposed in Appendix VII, where we evaluate the bosonic MDF at $10\tau$ in the anharmonic case, showing that it is significantly different from the quasi-particle MDF.

## 10 Integrability breaking: GHD and a collision term

In the quantum Newton's cradle experiment, there are numerous sources of integrability breaking. These include heating and losses, the presence of a trap, interactions between neighbouring tubes as well as hopping events between transverse levels of a given tube [49–51].

We can estimate the timescales for these processes in a variety of ways. For integrability breaking due to a trap, an extensive analysis has been provided in [42] for the hard rod gas. Here, for simplicity, we provide an estimate for the timescale using the notion of energy loss due to a "quantum jump" in our MD simulations. We elaborate on this in Appendix VIII A.

To estimate the timescales associated with other processes, we exploit the fact that GHD can readily take into account at least some integrability breaking processes through the addition of a collision term to the GHD equations. With such a collision term, the GHD equations, in terms of the space-time dependent filling fraction $n(\theta)$ (the GHD normal coordinates) [18, 19], read

$$\partial_t n(\theta) + v^{\text{eff}} \partial_x n(\theta) = f_{\text{collision}}(\theta). \tag{8}$$

In lowest order perturbation theory, $f_{\text{collision}}$ can be computed in terms of the exact matrix elements of the Lieb-Liniger model, for instance using the density matrix elements computed in [52]. As a simple demonstration, we sketch in Appendix VIII B $f_{\text{collision}}$ due to intertube interactions. We show that for the parameters governing the original quantum Newton's cradle experiment [1], $f_{\text{collision}}$ leads to changes in the energy at roughly the same rate as our estimate for the timescale of integrability breaking due to the trap.

## 11 Conclusion

The recently discovered Generalized HydroDynamics (GHD) is an ideal tool in order to provide a full account of the quantum Newton cradle experiment within the Lieb-Liniger model at the Euler scale, fully in the interacting regime. We have observed non-thermalization at cloud collisions, many-body elongation of the oscillation period, and many-body and single-body dephasing. We have also shown that GHD is the ideal tool to study trap expansions.

## Acknowledgements

We thank I. Bouchoule, N.J. van Druten, R. Dubessy, D. Gangardt, A. Minguzzi, M. Olshanii, S. Gopalakrishnan, and D. Weiss for helpful and stimulating discussions. We also thank R. Dubessy for useful comments on the manuscript. BD and JD thank the IESC Cargèse for hospitality during the school "Exact methods in Statistical Physics". BD also thanks the Perimeter Institute, Canada for hospitality where part of this work was done. TY thanks the Takenaka

Scholarship Foundation and the Tokyo Institute of Technology. RMK was supported by the U.S. Department of Energy, Office of Basic Energy Sciences, under Contract No. DE- AC02-98CH10886. J-SC acknowledges support from the European Research Council under ERC Advanced grant 743032 DYNAMINT.

## A  Double-well initial state

As a warm-up exercise, we have analyzed a setup that is a rather crude approximation of the QNC experiment [1], see Fig. 4. The initial state is the zero-temperature ground state of (1) in a double-well potential, which splits the gas into two well separated clouds. It is then evolved within a single harmonic well. The main phenomenon – the lack of thermalization upon cloud collisions – of the QNC experiment is observed. We compare both the zero-entropy GHD and molecular dynamics, with excellent agreement. In this setup, two important aspects of the QNC experiment are overlooked: the initial state is not at zero temperature, and the sequence of Bragg pulses produces an initial state with two sets of particles that are separated in momentum space rather than in real space. In the main text, we develop a more realistic approximation. The initial state is not a zero-entropy state, and we rely on the molecular dynamics simulation to solve the GHD equation.

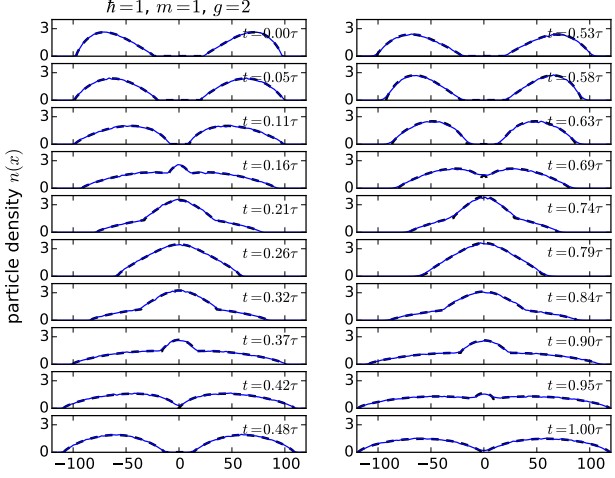

Figure 4: Lieb-Liniger gas at zero temperature released from a double-well potential to a harmonic trap $V(x) = m\omega^2 x^2/2$, on one full period $\tau = \frac{2\pi}{\omega}$, for parameters given in the text. Two methods for solving Eq. (3) are compared: zero-entropy GHD (dashed black curve), and the flea gas (blue).

We take $N = 250$ particles, and (in this paragraph only) we work in units where $\hbar = m = 1$ and $g = 2$. We take the initial state as the zero-temperature ground state of (1) in a double-well potential $V(x) = V_{\text{init}}(x) = 20\left((x/100)^4 - (x/100)^2\right)$, which splits the gas into two well separated clouds, each containing 125 particles. We construct the corresponding initial density of quasi-particles $\rho_{\text{p}}(\theta, x, 0)$ by searching for the zero-temperature hydrostatic solution of (3), equivalent to a local density approximation (LDA) [37]. Then, at time $t > 0$, the double-well is switched off and replaced by a harmonic trap $V(x) = m\omega^2 x^2/2 = x^2/800$. The GHD equation (3) is integrated using the second method —zero-entropy GHD— and third method —molecular dynamics— above. The results are presented in Fig. 4. The two methods are compared with perfect match: this is compelling evidence for the reliability and robustness of both methods for solving the GHD equation in a trap. Paralleling the QNC experiment, we

see that the two clouds initially propagate towards each other due to the harmonic potential, collide, and emerge to continue their quasi-harmonic motion. The interaction deforms the clouds substantially, but thermalization does not occur.

# B  Many-body dephasing and spreading in phase space

A free particle moving in a potential $V(x)$ preserves, at all times, its total energy $\theta^2/2 + V(x)$, where $\theta$ is its velocity (and here and below we take the particles' mass to be unity). Because of interactions, the particles of the Lieb-Liniger Bose gas of course do not preserve this energy. A related question is whether the quasi-particles of the GHD description of the gas do conserve it. We derive here explicitly the fact that the number $N(E)$ of particles within the energy region $\theta^2/2 + V(x) < E$ is not conserved (for any finite $E$), unless the effective velocity $v^{\mathrm{eff}}(\theta)$ equals the bare velocity $\theta$. Non-conservation is thus an effect of the interaction, and can be interpreted as a many-body dephasing effect. Conservation happens when the effective interaction is very weak: either in the Tonks-Girardeau limit, or in the free boson limit. In particular, the accuracy of the conservation of $N(E)$ – the distance between effective velocity and bare velocity – is a nontrivial (and nonlinear) function of the particle density. This helps explain the difference between the strength of the many-body dephasing effect in the harmonic and anharmonic cases, as explained in the main text.

Using $\theta_\pm(x) = \pm\sqrt{2(E - V(x))}$, we evaluate

$$
\begin{aligned}
\frac{d}{dt}N(E) &= \frac{d}{dt}\int dx \int_{\theta_-(x)}^{\theta_+(x)} d\theta\, \rho_{\mathrm{p}}(\theta, x) \\
&= -\int dx \int_{\theta_-(x)}^{\theta_+(x)} d\theta \left( \partial_x(v^{\mathrm{eff}}(\theta, x)\rho_{\mathrm{p}}(\theta, x)) - V'(x)\partial_\theta \rho_{\mathrm{p}}(\theta, x) \right).
\end{aligned}
\tag{9}
$$

Performing integration by part, the first term on the right-hand side gives

$$
\int dx \left( \partial_x \theta_+(x) v^{\mathrm{eff}}(\theta_+, x)\rho_{\mathrm{p}}(\theta_+, x) - \partial_x \theta_-(x) v^{\mathrm{eff}}(\theta_-, x)\rho_{\mathrm{p}}(\theta_-, x) \right)
\tag{10}
$$

and the second term

$$
\int dx\, V'(x)\left( \rho_{\mathrm{p}}(\theta_+, x) - \rho_{\mathrm{p}}(\theta_-, x) \right).
\tag{11}
$$

Clearly, $\partial_x \theta_\pm(x) = \mp V'(x)/\theta_\pm$. Thus we find

$$
\frac{d}{dt}N(E) = \int dx\, V'(x)\left[ \left(1 - \frac{v^{\mathrm{eff}}(\theta, x)}{\theta}\right)\rho_{\mathrm{p}}(\theta, x) \right]_{\theta_-(x)}^{\theta_+(x)}.
\tag{12}
$$

Thus the change of $N(E)$ is bounded by

$$
\int dx\, |V'(x)| \sum_{\pm} \left| 1 - \frac{v^{\mathrm{eff}}(\theta_\pm(x), x)}{\theta_\pm(x)} \right| \rho_{\mathrm{p}}(\theta_\pm(x), x),
\tag{13}
$$

which is controlled by the relative difference between $v^{\mathrm{eff}}(\theta, x)$ and $\theta$ at the region's boundaries $\theta_\pm(x)$.

A similar calculation gives the change of the energy within this region, $\mathcal{E}(E) = \int dx \int_{\theta_-(x)}^{\theta_+(x)} d\theta \, (\theta^2/2 + V(x)) \rho_{\mathrm{p}}(\theta, x)$,

$$\frac{d}{dt}\mathcal{E}(E) = \int dx \, V'(x) \left[ E \left( 1 - \frac{v^{\mathrm{eff}}(\theta, x)}{\theta} \right)_{\theta_-(x)}^{\theta_+(x)} + \int_{\theta_-(x)}^{\theta_+(x)} d\theta \, (v^{\mathrm{eff}}(\theta, x) - \theta) \rho_{\mathrm{p}}(\theta, x) \right]. \quad (14)$$

Further, by using the defining integral equation for the effective velocity, one has

$$\int_{-\infty}^{\infty} d\theta \, (v^{\mathrm{eff}}(\theta, x) - \theta) \rho_{\mathrm{p}}(\theta, x) = 0. \quad (15)$$

Therefore,

$$\frac{d}{dt}\mathcal{E}(E) = \int dx \, V'(x) \left[ E \left( 1 - \frac{v^{\mathrm{eff}}(\theta, x)}{\theta} \right)_{\theta_-(x)}^{\theta_+(x)} - \int_{\theta \notin [\theta_-(x), \theta_+(x)]} d\theta \, (v^{\mathrm{eff}}(\theta, x) - \theta) \rho_{\mathrm{p}}(\theta, x) \right]. \quad (16)$$

Again we see the same velocity difference playing an important role.

It is worth mentioning that for $E \to \infty$, we have $\theta_\pm(x) \to \pm\infty$, and recall that $v^{\mathrm{eff}}(\theta, x) \to \theta$ as $\theta \to \pm\infty$. In this limit it is clear that both $N(\infty)$ and $\mathcal{E}(\infty)$ are invariant, as they should as the system preserves the total number of particles and the total energy. It is also clear that in the free case, where $v^{\mathrm{eff}}(\theta, x) = \theta$, both quantities are preserved for all $E$s.

## C  A continuous family of conserved quantities for the GHD equation in a trap

Given a quasi-particle distribution function $\rho_{\mathrm{p}}(\theta)$, one defines the occupation number $n(\theta) = 2\pi \rho_{\mathrm{p}}(\theta)/1^{\mathrm{dr}}(\theta)$, where the dressing is defined as in the main text. The occupation number $n(\theta)$ is always between 0 and 1. As per the theory of GHD, this satisfies $\partial_t n + v^{\mathrm{eff}} \partial_x n - (\partial_x V/m) \partial_\theta n = 0$.

We find that, under GHD evolution in a trap —see Eq. (3) in the main text—

$$Q[f] := \int dx \, d\theta \, f(n(\theta, x, t)) \rho_{\mathrm{p}}(\theta, x, t) \quad (17)$$

is a conserved quantity for any function $f$, as long as the quasi-particle density $\rho_{\mathrm{p}}(\theta, x, t)$ does not have discontinuities in $\theta$ or $x$. To see this, notice that

$$\begin{aligned}
\partial_t(f(n)\rho_{\mathrm{p}}) &= f'(n)(\partial_t n)\rho_{\mathrm{p}} + f(n)\partial_t \rho_{\mathrm{p}} \\
&= f'(n)\left(-v^{\mathrm{eff}}\partial_x n + \frac{\partial_x V}{m}\partial_\theta n\right)\rho_{\mathrm{p}} + f(n)\left(-\partial_x(v^{\mathrm{eff}}\rho_{\mathrm{p}}) + \frac{\partial_x V}{m}\partial_\theta \rho_{\mathrm{p}}\right) \\
&= -\partial_x\left[v^{\mathrm{eff}}f(n)\rho_{\mathrm{p}}\right] + \partial_\theta\left[\frac{\partial_x V}{m}f(n)\rho_{\mathrm{p}}\right].
\end{aligned}$$

Upon integrating over $dx\,d\theta$, this gives zero using Stokes theorem, assuming zero quasi-particle density at infinity. Thus, $\partial_t Q[f] = 0$.

In fact, this can be understood as the generalization of the fact that the total Yang-Yang entropy is conserved in perfect fluids, hence in GHD. Namely, the total Yang-Yang entropy is

$$S_{\mathrm{YY}} = \int dx \, d\theta \, \frac{1}{n}(-n\log n - (1-n)\log(1-n))\rho_{\mathrm{p}}, \quad (18)$$

so it can be recast into the form of $Q[f]$ with $f(n) = \frac{1}{n}(-(1-n)\log(1-n)-n\log n)$. The precise structure of the infinite number of *dynamical* symmetries generated by these conserved quantities is still not known, and deserves further studies.

Finally, we note that a particularly convenient choice of basis for these conserved quantities corresponds to the choice $f(.) = \delta(.-\eta)$ for $\eta \in [0,1]$, thus leading to the conserved quantities

$$Q(\eta) := \int dx\,d\theta\,\delta(n(\theta,x,t)-\eta)\,\rho_{\mathrm{p}}(\theta,x,t) \tag{19}$$

mentioned in the main text. $Q(\eta)d\eta$ is the number of quasi-particles whose local occupation number lies between $\eta$ and $\eta + d\eta$.

# D   Coarse-graining and numerical simulations

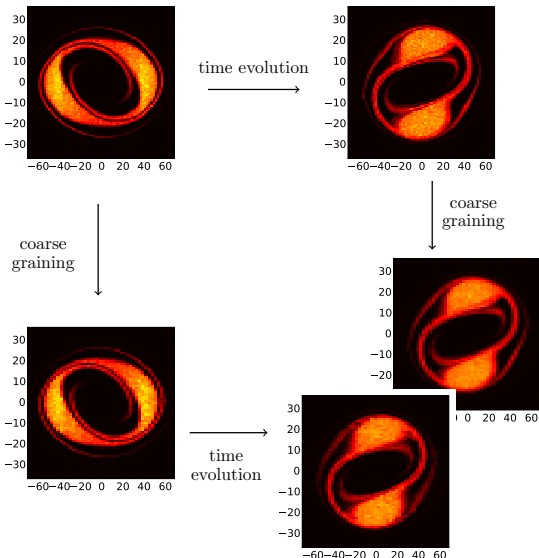

Figure 5: Diagram illustrating the fact that coarse-graining commutes with GHD time evolution. Starting from the state at $t = 8\tau$ in the anharmonic case (same data as in Fig. 2 in the main text), we first coarse-grain the system (i.e. we chose larger bins, and re-sample the phase-space distribution $\rho_{\mathrm{p}}(\theta,x)$ in our molecular simulator accordingly) and let it evolve during a time $\Delta t = 2\tau$. We compare the resulting distribution with the one obtained from evolving the system first, and then coarse-graining it. The two resulting distribution are almost identical.

## D.1   Invariance of GHD under coarse-graining

We consider the equation

$$\partial_t \rho_{\mathrm{p}} + \partial_x(v^{\mathrm{eff}}\rho_{\mathrm{p}}) + a\partial_\theta \rho_{\mathrm{p}} = 0, \tag{20}$$

where $a$ is the acceleration. The proof below is specialized to the simple case of the Lieb-Liniger model, but it is straightforward to extend it to the general context of GHD.

Consider coarse graining GHD, with coarse cells of area $\ell \times \ell'$ in phase space. That is, denote

$$\int_{\mathcal{C}(\theta,x)} d\gamma\,dy = \frac{1}{\ell\ell'}\int_{\theta-\ell/2}^{\theta+\ell/2} d\gamma \int_{x-\ell'/2}^{x+\ell'/2} dy \tag{21}$$

and let

$$\bar{\rho}_{\mathrm{p}}(\theta, x) = \int_{\mathcal{C}(\theta, x)} d\gamma dy \, \rho_{\mathrm{p}}(\gamma, y). \tag{22}$$

We make the following assumptions: (a) the acceleration is essentially constant on the scale $\ell'$, and (b) the velocity $\theta$ and (c) the differential scattering phase $\varphi(\theta)$ are essentially constant on the scale $\ell$. We also assume one of the following: either (d) the rapidity integral of quasi-particle densities and currents, on scale $\ell$, are essentially constant on scale $\ell'$ in the position variable; or (e) cells are uncorrelated,

$$\frac{1}{\ell^2 \ell'} \int_{\alpha-\ell/2}^{\alpha+\ell/2} d\gamma \int_{\theta-\ell/2}^{\theta+\ell/2} d\gamma' \int_{x-\ell/2}^{x+\ell/2} dy \rho_{\mathrm{p}}(\gamma, y) v^{\mathrm{eff}}(\gamma, y) \rho_{\mathrm{p}}(\gamma', y) \tag{23}$$

$$\approx \int_{\mathcal{C}(\alpha, x)} d\gamma dy \rho_{\mathrm{p}}(\gamma, y) v^{\mathrm{eff}}(\gamma, y) \int_{\mathcal{C}(\theta, x)} d\gamma dy \rho_{\mathrm{p}}(\gamma, y).$$

From the evolution equation,

$$
\begin{aligned}
\partial_t \bar{\rho}_{\mathrm{p}}(\theta, x) &= -\int_{\mathcal{C}(\theta, x)} d\gamma dy \left( \partial_y (v^{\mathrm{eff}}(\gamma, y) \rho_{\mathrm{p}}(\gamma, y)) + a(y) \partial_\gamma (\rho_{\mathrm{p}}(\gamma, y)) \right) \\
&= -\partial_x \left( \int_{\mathcal{C}(\theta, x)} d\gamma dy \, v^{\mathrm{eff}}(\gamma, y) \rho_{\mathrm{p}}(\gamma, y) \right) - \partial_\theta \left( \int_{\mathcal{C}(\theta, x)} d\gamma dy \, a(y) \rho_{\mathrm{p}}(\gamma, y) \right).
\end{aligned} \tag{24}
$$

Assuming (a) we have

$$\partial_t \bar{\rho}_{\mathrm{p}}(\theta, x) \approx -\partial_x \left( \int_{\mathcal{C}(\theta, x)} d\gamma dy \, v^{\mathrm{eff}}(\gamma, y) \rho_{\mathrm{p}}(\gamma, y) \right) - a(x) \partial_\theta \bar{\rho}_{\mathrm{p}}(\theta, x). \tag{25}$$

The above can also be written in integral form, so that the derivation holds in the space of weak solutions as well. Now define

$$\bar{v}^{\mathrm{eff}}(\theta, x) = \frac{\int_{\mathcal{C}(\theta, x)} d\gamma dy \, v^{\mathrm{eff}}(\gamma, y) \rho_{\mathrm{p}}(\gamma, y)}{\int_{\mathcal{C}(\theta, x)} d\gamma dy \, \rho_{\mathrm{p}}(\gamma, y)}. \tag{26}$$

Then clearly

$$\int_{\mathcal{C}(\theta, x)} d\gamma dy \, (v^{\mathrm{eff}}(\gamma, y) - \bar{v}^{\mathrm{eff}}(\theta, x)) \rho_{\mathrm{p}}(\gamma, y) = 0. \tag{27}$$

Thus

$$\partial_t \bar{\rho}_{\mathrm{p}}(\theta, x) \approx -\partial_x (\bar{v}^{\mathrm{eff}}(\theta, x) \bar{\rho}_{\mathrm{p}}(\theta, x)) - a(x) \partial_\theta \bar{\rho}_{\mathrm{p}}(\theta, x). \tag{28}$$

We now derive the integral equation for $\bar{v}^{\mathrm{eff}}(\theta, x)$ that shows that it is determined by $\bar{\rho}_{\mathrm{p}}(\theta, x)$. Assuming (b) and (c), we have

$$
\begin{aligned}
\bar{v}^{\mathrm{eff}}(\theta) &= \frac{\int_{\mathcal{C}(\theta, x)} d\gamma dy \left( v^{\mathrm{gr}}(\gamma, y) + \int d\alpha \varphi(\gamma - \alpha) \rho_{\mathrm{p}}(\alpha, y) (v^{\mathrm{eff}}(\alpha, y) - v^{\mathrm{eff}}(\gamma, y)) \right) \rho_{\mathrm{p}}(\gamma, y)}{\int_{\mathcal{C}(\theta, x)} d\gamma dy \, \rho_{\mathrm{p}}(\gamma, y)} \\
&\approx v^{\mathrm{gr}}(\theta, x) + \\
&\quad \frac{\int d\alpha \varphi(\theta - \alpha) \int_{\mathcal{C}(\theta, x)} d\gamma dy \rho_{\mathrm{p}}(\alpha, y) v^{\mathrm{eff}}(\alpha, y) \rho_{\mathrm{p}}(\gamma, y)}{\int_{\mathcal{C}(\theta, x)} d\gamma dy \, \rho_{\mathrm{p}}(\gamma, y)} \\
&\quad - \frac{\int d\alpha \varphi(\theta - \alpha) \int_{\mathcal{C}(\theta, x)} d\gamma dy \rho_{\mathrm{p}}(\alpha, y) v^{\mathrm{eff}}(\gamma, y) \rho_{\mathrm{p}}(\gamma, y)}{\int_{\mathcal{C}(\theta, x)} d\gamma dy \, \rho_{\mathrm{p}}(\gamma, y)}.
\end{aligned} \tag{29}
$$

Also from (c) we find in general

$$\int d\alpha \varphi(\theta - \alpha) f(\alpha) = \int d\alpha \int_{\mathcal{C}(\alpha)} d\gamma' \varphi(\theta - \gamma') f(\gamma') \approx \int d\alpha \varphi(\theta - \alpha) \int_{\mathcal{C}(\alpha)} d\gamma' f(\gamma'), \quad (30)$$

where

$$\int_{\mathcal{C}(\theta)} d\gamma = \frac{1}{\ell} \int_{\theta-\ell/2}^{\theta+\ell/2} d\gamma. \quad (31)$$

This gives, for the second terms in (29),

$$\int d\alpha \varphi(\theta - \alpha) \int_{\mathcal{C}(\theta,x)} d\gamma dy \rho_{\mathrm{p}}(\alpha, y) v^{\mathrm{eff}}(\alpha, y) \rho_{\mathrm{p}}(\gamma, y)$$

$$\approx \int d\alpha \varphi(\theta - \alpha) \int_{\mathcal{C}(\theta,x)} d\gamma dy \int_{\mathcal{C}(\alpha)} d\gamma' \rho_{\mathrm{p}}(\gamma', y) v^{\mathrm{eff}}(\gamma', y) \rho_{\mathrm{p}}(\gamma, y)$$

and similarly for the third term.

Now, on the one hand, assuming (d), we have

$$\int_{\mathcal{C}(\alpha)} d\gamma \rho_{\mathrm{p}}(\gamma, y) \approx \int_{\mathcal{C}(\alpha)} d\gamma \rho_{\mathrm{p}}(\gamma, x) \approx \int_{\mathcal{C}(\alpha,x)} d\gamma dy \rho_{\mathrm{p}}(\gamma, y), \quad x \in [y - \ell', y + \ell']. \quad (32)$$

Therefore,

$$\int d\alpha \varphi(\theta - \alpha) \int_{\mathcal{C}(\theta,x)} d\gamma dy \rho_{\mathrm{p}}(\alpha, y) v^{\mathrm{eff}}(\gamma, y) \rho_{\mathrm{p}}(\gamma, y)$$

$$\approx \int d\alpha \varphi(\theta - \alpha) \int_{\mathcal{C}(\theta,x)} d\gamma' dy' \int_{\mathcal{C}(\alpha,x)} d\gamma dy \rho_{\mathrm{p}}(\gamma', y') v^{\mathrm{eff}}(\gamma', y') \rho_{\mathrm{p}}(\gamma, y) \quad (33)$$

and, similarly,

$$\int d\alpha \varphi(\theta - \alpha) \int_{\mathcal{C}(\theta,x)} d\gamma dy \rho_{\mathrm{p}}(\alpha, y) v^{\mathrm{eff}}(\gamma, y) \rho_{\mathrm{p}}(\gamma, y)$$

$$\approx \int d\alpha \varphi(\theta - \alpha) \int_{\mathcal{C}(\theta,x)} d\gamma' dy' \int_{\mathcal{C}(\alpha,x)} d\gamma dy \rho_{\mathrm{p}}(\gamma', y') v^{\mathrm{eff}}(\gamma, y) \rho_{\mathrm{p}}(\gamma, y). \quad (34)$$

Putting these together, we find

$$\bar{v}^{\mathrm{eff}}(\theta, x) \approx v^{\mathrm{gr}}(\theta) + \int d\alpha \, \varphi(\theta - \alpha) \bar{\rho}_{\mathrm{p}}(\alpha, x)(\bar{v}^{\mathrm{eff}}(\alpha, x) - \bar{v}^{\mathrm{eff}}(\theta, x)). \quad (35)$$

On the other hand, assuming (e) we directly obtain (33) and (34), and the result (35) follows again.

That is, we conclude that $\bar{v}^{\mathrm{eff}}(\theta, x)$ is the effective velocity associated to the coarse-grained density $\bar{v}^{\mathrm{eff}}(\theta, x) = v^{\mathrm{eff}}[\bar{\rho}_{\mathrm{p}}](\theta, x)$. This is the GGE equation of state leading to GHD, and thus the coarse-grained equation (28) is GHD again.

## D.2 Numerical analysis

Numerical simulations have been performed with the molecular dynamics simulator proposed in [35] (using a standard desktop computer, 3.8GHz, quad core). We have performed simulations with the exact parameters described in the main text (giving approx. 350 particles),

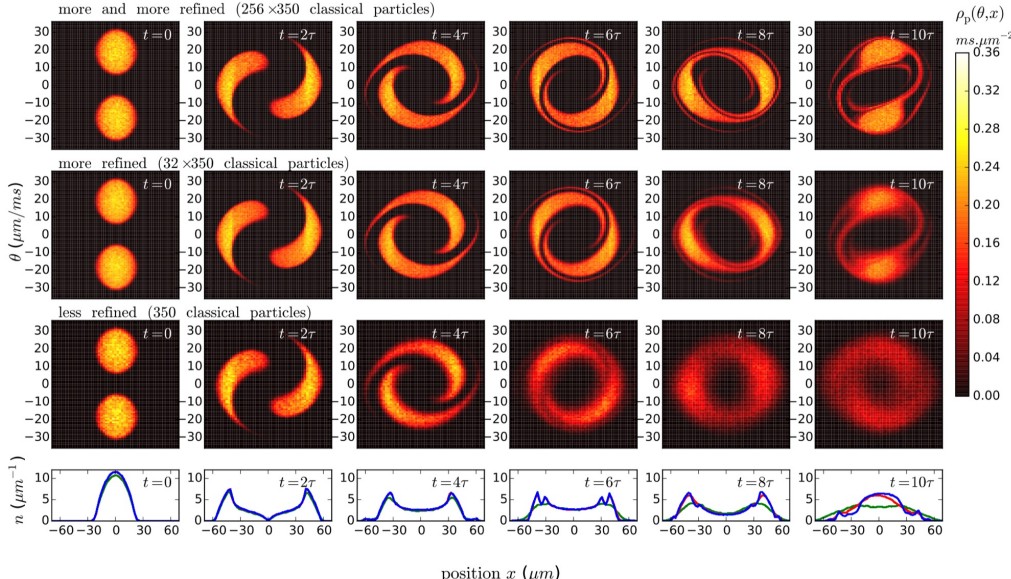

Figure 6: Comparison of the density $\rho_{\mathrm{p}}(\theta, x)$ obtained from three different microscopic realizations of the same GHD equation (anharmonic case): the lower one ("less refined") is our molecular simulator with 350 classical particles, the middle one ("more refined") is with $32 \times 350$ classical particles, and the top one ("more and more refined") is with $256 \times 350$ classical particles. The latter is the one shown in the main text in Figs. 1 and 2. The corresponding density profiles are plotted below (green: 350 class. part., red: $32 \times 350$ class. part., blue: $256 \times 350$ class. part.). As the discretization is refined, the density $\rho_{\mathrm{p}}(\theta, x)$ converges to the true GHD solution. A given discretization with a finite UV cutoff cannot resolve the fine structures that appear in GHD at scales smaller than the cutoff. However, GHD remains valid at larger scales, and this then amounts to coarse-graining.

as well as after rescaling all lengths by factors of $2^n$ for $n = 1, \ldots, 8$. Since GHD is manifestly invariant under scaling of lengths, these represent different choices of microscopy, with different numerical precision for the solution to the GHD equations. The equivalent of a sampling of 2000 has been used (that is, approx. $2000/2^n$ samples). In the harmonic case, this was observed to give a noise level (as calculated by the relative $L^1$ distance between two equivalent sampling) of the order of 5% throughout the evolution, on spectral densities binned on a $70 \times 70$ lattice covering the range of Figs. 1 and 2 (main text). In this case, we have found it sufficient to take $n = 4$ (approx. 5600 particles): we observed a Yang-Yang entropy production of approx. 6% over $10\tau$, and no significant change of the GHD-conserved function $Q(\eta)$ (defined in Section III).

The anharmonic case is much more delicate, and we have performed a more detailed numerical analysis. We show here results $n = 0$ (approx. 350 classical particles), $n = 5$ (approx. 11000 particles), and $n = 8$ (approx. 90000 particles; this with about half the sampling, using 4 samples only). The latter provides the results presented in the main text, see Fig. 2, reproduced (in the anharmonic case) in Fig. 6 for convenience, where we show the results at $t = 0, 2\tau, 4\tau, 6\tau, 8\tau, 10\tau$. It is apparent that agreement between the choices $n = 5$ and $n = 8$ is relatively good, although small-scale structures are more clearly discerned in the latter case. We have observed a Yang-Yang entropy production of approx. 20% over the evolution time of $10\tau$ for $n = 8$. More striking, however, is the analysis of the conserved quantity $Q(\eta)$. For the less refined discretization (350 classical particles), we see that it is conserved up to $4\tau - 7\tau$, then fine structures develop and are progressively erased by coarse

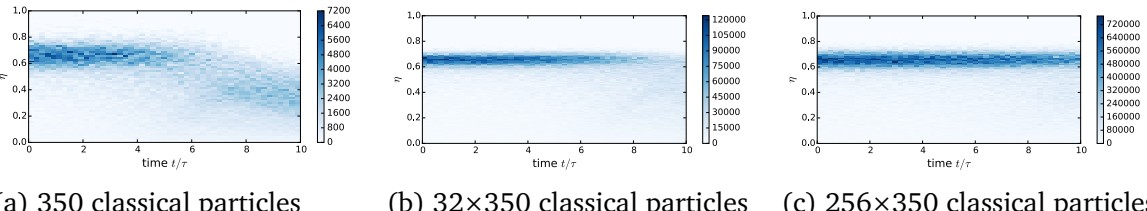

(a) 350 classical particles    (b) 32×350 classical particles    (c) 256×350 classical particles

Figure 7: Time-evolution of the quantities $Q(\eta)$, that are exactly conserved in pure GHD, but not in microscopic realizations of GHD. We see that, as the discretization is refined, and more and more fine structures of GHD are probed by the microscopic model, the quantities are conserved on longer times. Differences of thickness between the various cases are due to noise level differences.

graining, and after that $Q(\eta)$ is conserved again. For more refined discretizations, $Q(\eta)$ is conserved for a longer time, see Fig. 7. The distribution at $10\tau$ is relatively stable under change of the microscopy, as long as the number of particles is such that the corresponding coarse-graining is fine enough, in phase-space, for variations of the potential and scattering length to be small from cell to cell, yet large enough so that each cell contains a large number of particles (this happens to good approximation for $n \geq 6$ on a binning of $70 \times 70$, for instance). This lay support to the idea that coarse-grained GHD leads to the large-scale evolution, independently from the microscopy.

# E   Numerically obtained stationary state in the anharmonic case, and evidence that it is not thermal

We have investigated the stationary state obtained at large time. The setup is the same as in the main-text, however in order to speed up the many-body dephasing, we take a slightly stronger anharmonicity. We take the confining potential as $V(x) = (1+4(x/\ell)^2)\frac{m\omega^2 x^2}{2}$ with $\ell = 120\mu m$ and $\omega = 0.314 ms^{-1}$. We call $\tau = \frac{2\pi}{\omega}$. We observe that, at times than $t > 12\tau$, the distribution $\rho_p(\theta, x)$ looks stationary, see Fig. 8. We have compared this stationary distribution to the thermal distribution that has the same particle number and the same total energy. The two distributions are obviously different, as can be seen in the last plot of Fig. 8.

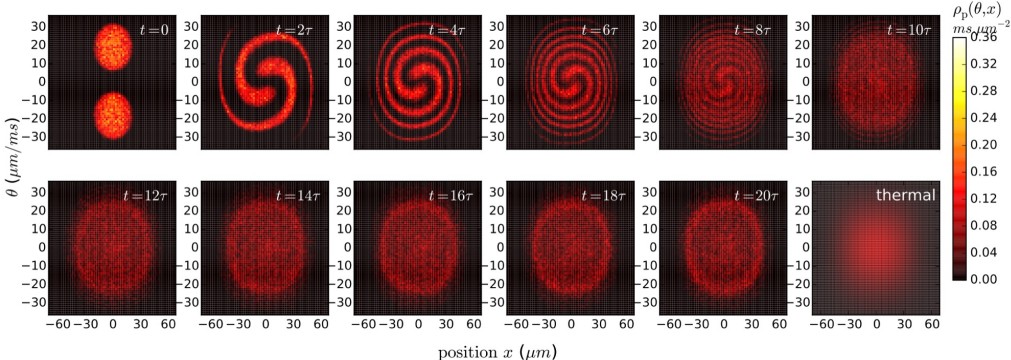

Figure 8: Comparison of the "stationary state" obtained numerically at $t = 20\tau$, and of the thermal distribution with the same number of particles and the same total energy.

## F  Trap release in 1d, and measurement of the momentum distribution of quasi-particles

Assume that we have a 1d Bose gas described, at some given time $t$ (which we fix to zero in this appendix), by a distribution of quasi-particles $\rho_{\rm p}(x,\theta)$, as in GHD. Assume that this density has a support that is contained in $[-\Delta x, \Delta x] \times [-\Delta\theta, \Delta\theta]$ so that no particle is outside the box $[-\Delta x, \Delta x]$, and no particle has a velocity larger than $\Delta\theta$. Then we release the longitudinal confinement, and let the gas expand in 1d. In this appendix, we are going to derive the following result: for a sufficiently long time of flight $T$, the spatial density of bosons $n(X,T) = \langle \Psi^\dagger(X)\Psi(X) \rangle$ is given by the momentum distribution function of the quasi-particles before the release, $n(\theta) = \int dx \rho_{\rm p}(x,\theta)$:

$$n(X,T) = \frac{1}{T}n(\theta), \qquad \text{with} \qquad \theta = X/T. \tag{36}$$

This result has long been known for the Tonks-Girardeau gas, where it has sometimes been dubbed "dynamical fermionization" [44, 45]. For the interacting case, it seems to have been pointed out only recently [46, 47]. Here, for the convenience of the reader, we provide a fully detailed proof of this result. The derivation of formula (36) consists of two steps.

The first step is to note that there must exist a time $T_1$ that is large enough such that the local density $n(x)$ is sufficiently low everywhere in the system, so that $\gamma(x) = c/n(x) \gg 1$ for all $x$. At that time $T_1$, the quasi-particle distribution is some function $\rho_{{\rm p},1}(x,\theta)$, with support in $[-\Delta x_1, \Delta x_1] \times [-\Delta\theta, \Delta\theta]$ for some $\Delta x_1$. Since, by construction in the Bethe ansatz method, quasiparticle spectral densities are exactly conserved under quantum evolution, we have

$$n(\theta) = \int dx \, \rho_{{\rm p},1}(x,\theta) = \int dx \, \rho_{\rm p}(x,\theta), \tag{37}$$

namely $n(\theta)$ was conserved during the evolution from $t = 0$ to $t = T_1$. Importantly, on the right-hand side, although the quantity $\rho_{\rm p}(x,\theta)$ is meaningfully defined only if the state is weakly varying in space (so that we can approximate it by a collection of homogeneous fluid cells), its spatial integral makes sense beyond this regime. Indeed, it simply encodes, as a function of $\theta$, the values of all extensive conserved quantities in the inhomogeneous initial state. Of course, in the application considered in the present work, $\rho_{\rm p}(x,\theta)$ is obtained after time evolution within an inhomogeneous external potential using the hydrodynamic approximation, and thus the values of all extensive conserved quantities it encodes are likewise subject to the hydrodynamic accuracy.

The second step goes as follows. At times $T > T_1$, because $\gamma$ is uniformly very large, the dynamics of the gas is captured by the Tonks-Girardeau hamiltonian,

$$H = \int dx \frac{1}{2m}(\partial_x \Psi_{\rm F}^\dagger)(\partial_x \Psi_{\rm F}), \tag{38}$$

where

$$\Psi_{\rm F}(x) = e^{i\pi \int_{u<x} du \, \Psi(u)\Psi(u)} \Psi(x). \tag{39}$$

Thus, we are back to the case of the Tonks-Girardeau gas, and we simply apply the results of [44, 45]. For completeness, here we give a fully detailed calculation that leads to the wanted result.

The density of bosons at point $X$ and time $T$ is

$$
\begin{aligned}
n(X,T) &= \left\langle \Psi^\dagger(X)\Psi(X)\right\rangle_T \\
&= \left\langle \Psi_F^\dagger(X)\Psi_F(X)\right\rangle_T \\
&= \int \frac{dk}{2\pi}\int \frac{dk'}{2\pi} e^{i(k-k')X-i(T-T_1)[\epsilon(k)-\epsilon(k')]}\left\langle \Psi_F^\dagger(k)\Psi_F(k')\right\rangle_{T_1} \\
&= \int \frac{dk}{2\pi}\int \frac{dk'}{2\pi} e^{i(k-k')X-i(T-T_1)[\epsilon(k)-\epsilon(k')]}\int dy\, e^{-i(k-k')y} \\
&\qquad \int \frac{dq}{2\pi} e^{iqy}\left\langle \Psi_F^\dagger(\frac{k+k'}{2}+\frac{q}{2})\Psi_F(\frac{k+k'}{2}-\frac{q}{2})\right\rangle_{T_1} \\
&= \int \frac{dk}{2\pi}\int \frac{dk'}{2\pi} e^{i(k-k')X-i(T-T_1)[\epsilon(k)-\epsilon(k')]}\int dy\, e^{-i(k-k')y}\frac{2\pi}{m}\rho_{p,1}(y,\frac{k+k'}{2m}).
\end{aligned}
\tag{40}
$$

In the last line, we used the fact that $\int \frac{dq}{2\pi}e^{iqx}\left\langle \Psi_F^\dagger(m\theta+\frac{q}{2})\Psi_F(m\theta-\frac{q}{2})\right\rangle_{T_1}$ is nothing but the Wigner function, so it is exactly the number of fermions at position $x$ with momentum $m\theta$, therefore it has to be equal to $\frac{2\pi}{m}\rho_{p,1}(x,\theta)$. [The factor $\frac{2\pi}{m}$ simply comes from a difference in normalization convention between $\rho_{p,1}$ for the Tonks-Girardeau gas and the Wigner function: for instance, the total number of particles is $\int dx d\theta \rho_{p,1} = \int \frac{dx d(m\theta)}{2\pi}W$, if $W$ is the Wigner function.]

Since $\epsilon(k)-\epsilon(k') = \frac{k^2}{2m}-\frac{k'^2}{2m} = \frac{(k+k')(k-k')}{2m}$, this gives (with the change of variables $K=\frac{k+k'}{2}$, $q=k-k'$):

$$
\begin{aligned}
n(X,T) &= \int dy \int \frac{dK}{m}\int \frac{dq}{2\pi} e^{-iq[y-X+\frac{K}{m}(T-T_1)]}\rho_{p,1}(y,\frac{K}{m}) \\
&= \int \frac{dK}{m}\rho_{p,1}\left(X-\frac{K}{m}(T-T_1),\frac{K}{m}\right).
\end{aligned}
\tag{41}
$$

This is essentially the result we want: it expresses the fact that the number of bosons at position $X$ at time $T$ is the one of fermions at time $T=T_1$ that have traveled a distance $K/m(T-T_1)$, so they must have velocity $K/m$. Notice that the statistics of the particles does not play a role in the argument, and even though we are doing calculations with the fermions, in the end we have a result valid for the density of bosons.

Finally, to get the more compact formula (36), we use the fact that $\rho_{p,1}(x,\theta)$ is zero if $x\notin[-\Delta x_1,\Delta x_1]$, so

$$
\int_{-\infty}^{\infty}\frac{dK}{m}\rho_{p,1}\left(X-\frac{K}{m}(T-T_1),\frac{K}{m}\right) = \int_{m\frac{X-\Delta x_1}{T-T_1}}^{m\frac{X+\Delta x_1}{T-T_1}}\frac{dK}{m}\rho_{p,1}\left(X-\frac{K}{m}(T-T_1),\frac{K}{m}\right),
$$

with an integrant centered around $m\Theta$, where $\Theta = X/(T-T_1)$. Taking $T-T_1\gg m\Delta x_1$, we can substitute the second argument of $\rho_{p,1}$,

$$
\int_{-\infty}^{\infty}\frac{dK}{m}\rho_{p,1}\left(X-\frac{K}{m}(T-T_1),\frac{K}{m}\right) \underset{T-T_1\gg m\Delta x_1}{=} \int_{m\frac{X-\Delta x_1}{T-T_1}}^{m\frac{X+\Delta x_1}{T-T_1}}\frac{dK}{m}\rho_{p,1}\left(X-\frac{K}{m}(T-T_1),\Theta\right)
$$

$$
= \frac{1}{T-T_1}\int_{-\Delta x_1}^{\Delta x_1}du\,\rho_{p,1}(u,\Theta),
$$

where we have set $u=X-K/m(T-T_1)$. We thus arrive at

$$
n(X,T) = \frac{1}{T-T_1}n(\Theta), \qquad \text{with} \qquad \Theta = X/(T-T_1).
\tag{42}
$$

If we further assume that $T\gg T_1$, then we get Eq. (36) as claimed.

# G   Computing the bosonic MDF

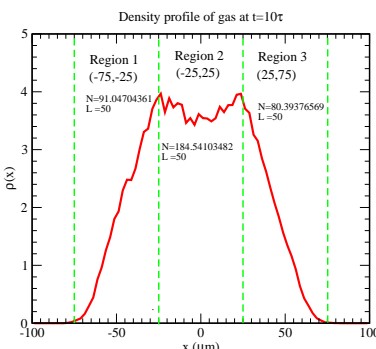
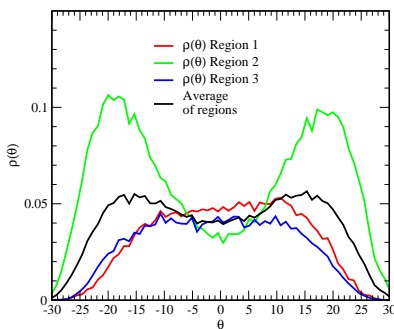

Figure 9: Left: Density profile of the gas at time $10\tau$ as obtained through GHD. For practical computations of the MDF, three regions are defined, each with characteristic root density function (see text). Right: Root density functions for each of the three regions defined in the left panel. Each curve is normalized to unit filling for convenience.

The starting point for the calculation of the bosonic momentum distribution function in the anharmonic trap is the full spatial density profile of the gas at time $t = 10\tau$ obtained from GHD and plotted in the left panel of Fig. 9. This density is subsequently divided into three separate regions, as illustrated again in the left panel of Fig. 9. For each region, a root density $\rho_i(\theta)$ is extracted. These are plotted on the right panel of Fig. 9. They satisfy the sum rule

$$\int d\theta \rho_i(\theta) = \frac{N_i}{L_i} = n_i.$$

These distributions clearly show that the gas in each of the three regions is significantly excited away from the ground state.

The next step is to compute the bosonic MDF separately on each of these three individual constituent representative states. To do this, we rescale the $\rho_i(\theta)$'s via

$$\tilde{\rho}(\theta) = \rho_i(n_i\theta), \tag{43}$$

so that we are working at unit density, i.e.

$$\int d\theta \tilde{\rho}_i(\theta) = 1.$$

We then use ABACUS [13–17] to compute the MDF on each representative state. This involves the following steps: starting from each individual $\tilde{\rho}_i(\theta)$, a best-fitting discretized Bethe state $|i\rangle_N$ is constructed at a chosen particle number $N_{ABACUS}$ (setting this equal to $L_{ABACUS}$ to stay at unit filling) by choosing a set of quantum numbers generated from the state's counting function, namely: adding a rapidity whenever $L \int_{-\infty}^{\lambda} d\lambda' \rho_i(\lambda')$ crosses a half-odd integer, and setting the quantum numbers to those giving the closest-matching set of rapidities; N is chosen even, and as large as practically possible. ABACUS is then run for the one-body correlation

$_N\langle i|\psi^\dagger(x,t)\psi(0,0)|i\rangle_N$. The quality of the result is quantified by the saturation of the integrated intensity sum rule. On such highly-excited states, a large number of intermediate states must be summed over (for $N_{ABACUS} = 32$, these were 74307322 ($i = 1$), 101334549 ($i = 2$) and 87195380 ($i = 3$), yielding saturations of 0.964412, 0.933457 and 0.979004).

Having these three MDFs ($\tilde{n}_i(k), i = 1, 2, 3$), we now rescale back to MDFs ($n_i(k), i = 1, 2, 3$) corresponding to the original three regions characterized by $N_i, L_i$. The relevant relation here is

$$n_i(k, N_i, L_i)) = c_i(N_i, L_i)\tilde{n}_i(kL_{ABACUS}/L, N_{ABACUS}, L_{ABACUS}) \tag{44}$$

Here $c_i$ is a constant that can be determined by insisting that

$$\int dk n_i(k, N_i, L_i) = n_i.$$

The results are displayed in Fig. 10. We then average over the three $n_i(k)$ to obtain what would be the MDF measured in the actual experiment. If we compare the r.h.s. of Fig.9 and Fig. 10, we see the bosonic MDF and $\rho(\theta)$ are considerably different. However both have a double humped feature characteristic of post-Bragg pulse states.

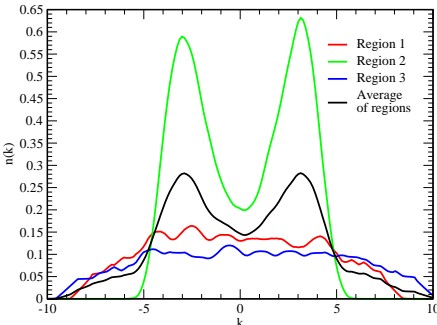

Figure 10: Bosonic momentum distribution function of the trapped gas at $t = 10\tau$, computed through three representative states using ABACUS (see text). The characteristic double-peak structure of the post-Bragg pulse state is clearly seen.

# H    Timescales for Integrability Breaking

We consider in this section estimates for timescales due to integrability breaking. We do this in two parts. In the first part we consider the timescale associated with integrability breaking due to the trap. And in the second part, we consider the derivation of a GHD collision term coming from intertube interactions.

## H.1    Estimate of Integrability Breaking due to Trap

We have argued in the main text that integrability breaking due to the trapping potential is small. We provide here an argument for this [2].

---

[2]We are indebted to Igor Mazets for suggesting this line of argument to us.

One way to parameterize the integrability breaking is to consider how the potential energy of a particle changes when two particles collide in the presence of the trap. We can estimate this energy by using the molecular dynamics representation of the GHD equations. In the molecular dynamics, when two particles collide, they experience a quantum jump, $\Delta x$, where the particles are displaced according to their relative momentum, $p_1 - p_2$, and the strength of interactions:

$$\Delta x = \frac{2c}{((p_1 - p_2)/\hbar)^2 + c^2}.\tag{45}$$

The quantum displacement gives us a scale for integrability breaking because upon displacement in the molecular dynamics simulation, the potential energy of a particle experiencing a quantum displacement changes by an amount (so violating energy conservation):

$$\Delta V \sim \frac{dV_{\text{trap}}}{dx}\Delta x \sim m\omega^2 L\,\Delta x.\tag{46}$$

Rather than trying to estimate this directly (it is difficult to provide even a back of the envelope estimate of this quantity as it requires accounting for both inter- and intra-cloud collisions), we read off the change of energy directly from our numerical simulations. We find it to be approximately 0.1% for each oscillation of the clouds. This rate is smaller than what we estimate in the next section for the intertube interactions present in the QNC experiment.

## H.2 Collision Term due to Density-Density Couplings Between Tubes

In the main text we have discussed the possibility of adding a collision term to the GHD equations. Here we elaborate on how to compute the collision term and from it provide an estimate of the time scale for integrability breaking.

For this exercise, we are going to consider a system composed of two Lieb-Liniger models that are coupled by a density-density interaction. As we will be deriving the collision term in lowest order perturbation theory, the effects of having more tubes coupling to one another, either because the tubes are in an array of a given coordination number (as is typical) or because of long range dipolar forces (as in Ref. [49]) are additive. The case of two coupled Lieb-Liniger models is then sufficiently general.

The Hamiltonian we will then consider

$$
\begin{aligned}
H_{\text{pert}} &= A\int_0^R dx\,\rho_1(x)\rho_2(x);\\
&= AL\sum_k \rho_{1k}\rho_{2-k},
\end{aligned}\tag{47}
$$

where $\rho_{i,k}$, the Fourier component of the density operator in the i-th tube, is defined as

$$\rho_{i,k} = \frac{1}{L}\sum_q \psi_{i,k+q}^\dagger \psi_{i,q},\tag{48}$$

where $\psi_{i,q}$ is the $q$-th Fourier component of the $i$-th chain field operator.

To compute the collision term, i.e. the rate of change of the quantum numbers $I_r$ in a state, we imagine that we have an initial state $|i\rangle$, characterized by a set of occupied quantum numbers

$$\{I_{r,s}\}_{r=1}^N,\quad s=1,2.\tag{49}$$

These integers $I_{r,s}$ are the the Bethe integers for which one solves the Bethe ansatz equations describing the uncoupled chains.

Now let $n_s(I_{r,s})$ be the occupation of quantum number $I_{r,s}$ on chain $s$. By the Fermi golden rule, the rate of change of $n_s(I_{r,s})$ from its initial value is

$$
\begin{aligned}
f_{\text{collision}}(I_{r,s}) & \equiv \dot{n}_s(I_{r,s}) \\
& = \sum_f R_{fi}\Big[ n_{f,s}(I_{r,s})(1 - n_{i,s}(I_{r,s})) - n_{i,s}(I_{r,s})(1 - n_{f,s}(I_{r,s})) \Big] \\
& = \sum_f R_{fi}\Big[ n_{f,s}(I_{r,s}) - n_{i,s}(I_{r,s}) \Big],
\end{aligned}
\tag{50}
$$

where we are summing over all final states, $f$, and

$$
\begin{aligned}
R_{fi} &= 2\pi |\langle f|H_{\text{pert}}|i\rangle|^2 \delta(\omega_f - \omega_i), \\
n_{f,s}(I_{r,s}) &= \text{occupation of } I_{r,s} \text{ on chain s in the final state } f, \\
n_{i,s}(I_{r,s}) &= \text{occupation of } I_{r,s} \text{ on chain s in the initial state i.}
\end{aligned}
\tag{51}
$$

To develop this expression for $\dot{n}_s(I_{r,s})$ further, we write the states $|i\rangle, |f\rangle$ explicitly as a product state of states belonging to the two chains:

$$
|i\rangle = |i_1\rangle |i_2\rangle; \quad |f\rangle = |f_1\rangle |f_2\rangle.
\tag{52}
$$

We, for simplicity, will take the initial states on the two chains to be equal, i.e. $|i_1\rangle = |i_2\rangle$. We will also only consider the first set of final states that can lead to thermalization. Such states involve 2-particle-hole excitations on one chain, and 1-particle hole excitation on the other, i.e. $|f_{s=1,2}\rangle$ are given by

$$
|f_1\rangle = |i_1, \hat{h}_1, \hat{h}'_1, p_1, p'_1\rangle, \quad |f_2\rangle = |i_2, \hat{h}_2, p_2\rangle,
\tag{53}
$$

where here a state

$$
|i_s, \hat{h}_s, \hat{h}'_s \cdots, p_s, p'_s, \cdots\rangle
\tag{54}
$$

is defined as the state $|i_s\rangle$ on chain $s$ with quantum numbers (holes) $h_s, h'_s, \cdots$ removed and quantum numbers (particles) $p_s, p'_s, \cdots$ added. States involving only 1-particle-hole excitation on each chain can lead to equilibriation between the chains, but will not thermalize non-Gibbsian distributions. As such, these final states will not be considered.

With these assumptions, we can write the matrix element square, $R_{fi}$, as

$$
\begin{aligned}
R_{fi} = {}& \frac{32\pi A^2}{L^2} \delta(\omega_f - \omega_i)\, \delta_{p_1+p'_1+p_2-h_1-h'_1-h_2,0}\, F^2_{p_1,p'_1,h_1,h'_1} F^2_{p_2,h_2} \\
& \times n_i(h_1) n_i(h'_1) n_i(h_2)(1 - n_i(p_1))(1 - n_i(p'_1))(1 - n_i(p_2)) + \big(1 \leftrightarrow 2\big),
\end{aligned}
\tag{55}
$$

where $n_i(I) = 0, 1$ marks the presence or absence of the quantum number $I$ in the initial state $|i_s\rangle$, and $F_{p_1,\cdots,h_1,\cdots}$ is the matrix element for the density operator on one of the chains involving particles $p_1, \cdots$ and holes $h_1, \cdots$. The occupation of the quantum number $I_{r,s}$ in the final state on chain $s$ is given by

$$
n_{f,s}(I_{r,s}) = \Big(\prod_{h_s} n_i(h_s)\Big)\Big(\prod_{p_s}(1 - n_i(p_s))\Big)\Big(n_i(I_{r,s}) - \sum_{h_s} \delta_{I_{r,s},h_s} + \sum_{p_s} \delta_{I_{r,s},p_s}\Big).
\tag{56}
$$

We are then able to write the rate of change of quantum numbers $n_{I_1}$ (where we take chain 1 for specificity) as follows

$$
\begin{aligned}
\dot{n}_1(I_{r,1}) \;=\; & \frac{8A^2}{\pi} \sum_{h_1,h_1',h_2,p_1,p_1',p_2} \Bigg[ (F_{p_1,p_1',h_1,h_1'} F_{p_2,h_2})^2 \\
& \delta(p_1{}^2 + p_1'{}^2 + p_2{}^2 - h_1{}^2 - h_1'{}^2 - h_2{}^2)\delta_{p_1+p_1'+p_2-h_1-h_1'-h_2,0} n_i(h_1) n_i(h_1') n_i(h_2) \\
& \times (1 - n_i(p_1))(1 - n_i(p_1'))(1 - n_i(p_2))(\delta_{I_{r,1},p_1} + \delta_{I_{r,1},p_1'} - \delta_{I_{r,1},h_1} - \delta_{I_{r,1},h_1'}) \Bigg] \\
& + \frac{8A^2}{\pi} \sum_{h_1,h_2,h_2',p_1,p_2,p_2'} \Bigg[ (F_{p_2,p_2',h_2,h_2'} F_{p_1,h_1})^2 \\
& \times \delta(p_1^2 + p_2^2 + p_2'^2 - h_1^2 - h_2^2 - h_2'^2)\delta_{p_1+p_2+p_2'-h_1-h_2-h_2',0} n_i(h_1) n_i(h_2) n_i(h_2') \\
& \times (1 - n_i(p_1))(1 - n_i(p_2))(1 - n_i(p_2'))(\delta_{I_{r,1},p_1} - \delta_{I_{r,1},h_1}) \Bigg].
\end{aligned}
\tag{57}
$$

Here the first term in the above corresponds to the case where the 2-particle-hole excitation takes place on chain 1 and the 1-particle-hole excitation on chain 2 while the second term exchanges the chains where these two processes occur.

We are not going to evaluate this expression in detail as the 2-particle hole matrix elements $F_{p_1,p_1',h_1,h_1'}$ are highly non-trivial. We can however provide an estimate of the time scale. We know that matrix elements $F_{p_1,p_1',h_1,h_1'}$ scale as $1/c$ (and so in the $c = \infty$ limit this process would be suppressed and thermalization would not occur). In general, density matrix elements involving n-particles and n-holes scale as $1/c^{n-1}$. The contribution then from the above sum goes as $N^3/c^2$ ($N$ is the number of particles) with the result

$$
\dot{n}_1(I_1) \sim A^2 N \frac{n^2}{c^2} \frac{\hbar}{m},
\tag{58}
$$

where here $m$ is the mass of rubidium atom. We can write the density-density coefficient $A$ coupling the tubes together as $\gamma_{intertube}\rho_0$ where $\rho_0$ is the background density in the tubes.

$$
\dot{n}_1(I_1) \sim N \frac{\hbar}{m} n^2 \frac{\gamma_{intertube}^2}{\gamma_{intratube}^2},
\tag{59}
$$

This then implies the change in energy of the gas due to intertube interactions is

$$
\dot{E} \sim N \frac{\hbar^3 n^4 \gamma_{intertube}^2}{m^2 \gamma_{intratube}^2}.
\tag{60}
$$

As the energy goes as $E \sim N\hbar^2 n^2/m$ we see that

$$
\frac{\dot{E}}{E} \sim \frac{\hbar n^2 \gamma_{intertube}^2}{m \gamma_{intratube}^2} \sim 10^4 s^{-1} \frac{\gamma_{intertube}^2}{\gamma_{intratube}^2}.
\tag{61}
$$

We know in the context of Ref. [1] that $\gamma_{intertube} \ll \gamma_{intratube}$. If, for example, $\gamma_{intratube} \sim 10^{-2}$, we see that the energy change per oscillation of the gas is on the order of 1%, similar to our estimates of the energy change due to integrability breaking arising from the trap. However this rough estimate shows that if $\gamma_{intratube} \sim \gamma_{intertube}$ as in Ref. [49], the fractional change in energy per oscillation cycle will be $\mathcal{O}(1)$.

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
