# Peer review of "Hydrodynamics of the interacting Bose gas in the Quantum Newton Cradle setup"

_SciPost Physics, doi:SciPost Phys. 6, 070 (2019)_

## Round 2 · Referee Report · Anonymous · 2018-12-14

Strengths

1. Well formulatied physical questions and motivation
2. Relation to real experiment.
3. Elegant method
4. Convincing results
5. Beautiful figures

Weaknesses

1. Absence of discussion of integrability breaking mechanisms (see report).

Report

The manuscript uses Generalized Hydrodynamics (GHD) to describe time evolution
of interacting bosons in one dimension confined by harmonic potential
following the groundbreaking Newton Cradle experiment. The main results are
accurate predictions for the density profiles and quasi-momentum distribution
which can be in principle observed using one-dimensional time of flight. The
results are consistent with the absence of thermalization after several
thousands of oscillations in the trap observed in the original experiment.

The paper provides a valuable application of Generalized Hydrodynamics to real
experiments and illustrates its predictive power. The authors give detailed
account of the method and its limitations. In particular they are able to
show how a smooth phase-space distributions emerge from coarse graining.

My main question is whether the confinement is the main source of the
integrability breaking in this system? Is it possible that other sources of
non-integrability, like three body interactions due to transitions to higher
transverse states, will lead to significant deviations of the idealized picture
and whether they can be accounted for by suitable modifications of Generalized
Hydrodynamic equations? Such discussion can be very valuable for realistic
experiments.

The manuscript is of clear interest to the theoretical community studying
dynamics of quantum systems close to integrability and the presented method
can be used for efficient simulations of future experiments with ultra cold
gases.

Requested changes

No specific changes are required to the current version.

---

## Round 2 · Referee Report · Anonymous · 2018-12-31

Strengths

1. Applies generalized hydrodynamic (GHD) approach to integrable models in an external potential to experimentally relevant (Newton's Cradle) setup.

2. Method is not limited to weak interaction or short times.

3. Investigates the effect of interaction and trap anharmonicity on dephasing.

4. Identifies infinite family of conserved quantities of GHD that constrains thermalization

Weaknesses

Only minor weaknesses, identified in the "requested changes"

Report

This paper represents an important part of a series of theoretical developments sparked by the Quantum Newton's Cradle experiment of 2006. The generalized hydrodynamic (GHD) approach to integrable models developed in earlier papers by the same authors and others. The important aspect of the present work is the analysis of realistic conditions that resemble the initial experiment, namely the presence of a trap potential, including anharmonic contributions.

The manuscript itself is extremely clearly written, with the background theory clearly summarized before the numerical simulations of the GHD equations are presented. The various features of the solutions are then discussed, particularly in light of the original experiment.

Requested changes

1. Clarification of the "Euler scale" in the Section "The GHD equation". What is it? The description is purely verbal, and says that the variation of conserved densities is "slow enough". Compared to what? Usually one justifies such an approach via local equilibration, but that is not happening here.

2. "In standard approaches, this would imply....". I see by the citation that the authors refer to application in BEC. In general it isn't true that the distribution function in a nonequilibrium situation is a boosted Gibbs state. If it were, the collision integral in the Boltzmann equation would vanish!

3. I believe the numerical technique used here goes by the name "Smoothed particle hydrodynamics" elsewhere. If the authors agree it may be worth mentioning this.

4. The caption to Fig. 1 contains the term "Flea gas" which I believe is introduced in other work by the authors, but not here. It should be explained or removed.

5. End of the section on harmonic QNC: "...for entropic reasons the particle density spreads mostly towards lower energies...". Can some words of explanation be added here?

6. Some explanation for the chosen anharmonicity could be provided i.e. why not a quartic potential?

---

## Round 3 · Author Response

We thank the Referees for their work on our manuscript. We provide a detailed reply to the comments below.

Report 2

"This paper represents an important part of a series of theoretical developments sparked by the Quantum Newton's Cradle experiment of 2006. The generalized hydrodynamic (GHD) approach to integrable models developed in earlier papers by the same authors and others. The important aspect of the present work is the analysis of realistic conditions that resemble the initial experiment, namely the presence of a trap potential, including anharmonic contributions.

The manuscript itself is extremely clearly written, with the background theory clearly summarized before the numerical simulations of the GHD equations are presented. The various features of the solutions are then discussed, particularly in light of the original experiment. Requested changes

  1. Clarification of the "Euler scale" in the Section "The GHD equation". What is it? The description is purely verbal, and says that the variation of conserved densities is "slow enough". Compared to what? Usually one justifies such an approach via local equilibration, but that is not happening here."

Answer : The Euler scale refers the largest possible length- and time-scales at which the system is described. It is the scale at which no dissipation occurs. At that scale, the fluid cells are viewed as thermodynamically large local subsystems, each of which is in a translation-invariant stationary state which maximises entropy. Since each fluid cell is viewed as translation-invariant, the currents associated to the conserved quantities depend only on the densities of charges inside a fluid cell (not on their derivatives). Because the currents do not depend on the derivatives, there is no dissipation.

In order for an Euler-scale description to be valid, the variation of the charge densities must be slow compared to all microscopic scales (e.g. the interparticle distance). The Euler scale is always the zeroth order of any hydrodynamic description, and it is a concept which is used in many references/books ; we added reference [20] there for clarity.

"Local equilibration", although commonly used as a phrase, is not quite the right concept / description of the idea of the Euler scale in general. "Local stationarity" is a better way of describing the Euler scale, or "local entropy maximisation", where entropy maximisation is with respect to all available conserved quantities.

"2. "In standard approaches, this would imply....". I see by the citation that the authors refer to application in BEC. In general it isn't true that the distribution function in a nonequilibrium situation is a boosted Gibbs state. If it were, the collision integral in the Boltzmann equation would vanish!"

Answer: The references are works which used the standard Euler hydrodynamics in order to describe cold atomic gases. In standard hydrodynamics, the fact that the system is locally in a boosted Gibbs state is the direct consequence of local equilibration and Galilean invariance.

This would be correct in a non-integrable Galilean gas described at the Euler scale. But here the main point is that this is incorrect for the integrable system describing 1d Bose gases."

Our remark about « Standard approaches...»  does not refer to the Boltzmann equation. The Boltzmann equation describes non-equilibrium situations for non-integrable gases, which are beyond the Euler scale. At the Euler scale, those gases would simply be assumed to be locally at equilibrium ; on the contrary, the Boltzmann equation describes the evolution towards local equilibrium, which is happening on much shorter time-scales.

"3. I believe the numerical technique used here goes by the name "Smoothed particle hydrodynamics" elsewhere. If the authors agree it may be worth mentioning this."

Answer: we were not aware of the SPH methods. Having looked quickly, it seems related, but it is not clear to us that the technically precise family of algorithms SPH refers to (in particular with the kernel function, specific approximations for gradients, etc.) is really describing what we are using. In fact, what we are using might be closer to what people refer to as molecular dynamics, and a generalisation of the hard rod gas [26], as explained in [24]. In any case, we prefer not making too strong statements as to how the method connects with other numerical methods used in fluid dynamics, in order to avoid any confusion.

"4. The caption to Fig. 1 contains the term "Flea gas" which I believe is introduced in other work by the authors, but not here. It should be explained or removed."

Answer: thank you for noting this; we added the reference [24].

"5. End of the section on harmonic QNC: "...for entropic reasons the particle density spreads mostly towards lower energies...". Can some words of explanation be added here?"

Answer: the graphs are results of numerics, and the phrase highlighted was an interpretation of the shape seen in the harmonic case from general physical intuition. Here the intuition stems simply from the fact that spreading occurs because of interaction, and that total energy is conserved. Spreading must therefore be stronger at lower energies than at higher energies because (either by the form of the potential, or the form of the kinetic energy as function of momentum), the change in energy per distance (per momentum) is greater at higher energies than it is at lower energies. Perhaps "entropic principles" was not quite the right phrase (at Euler scale, no fluid-cell entropy is produced), so we have re-phrased this.

"6. Some explanation for the chosen anharmonicity could be provided i.e. why not a quartic potential?"

Answer: we were attempting to be closer to the actual experimental setup, where potentials are closer to trigonometric functions, which is what we chose. In particular, the anharmonicity has the property that the potential is smaller than harmonic at further away from the centre (i.e. becomes flatter), different from a simple x^4 addition. We added comments on page 4, left on this.

Report 1

"The manuscript uses Generalized Hydrodynamics (GHD) to describe time evolution of interacting bosons in one dimension confined by harmonic potential following the groundbreaking Newton Cradle experiment. The main results are accurate predictions for the density profiles and quasi-momentum distribution which can be in principle observed using one-dimensional time of flight. The results are consistent with the absence of thermalization after several thousands of oscillations in the trap observed in the original experiment.

The paper provides a valuable application of Generalized Hydrodynamics to real experiments and illustrates its predictive power. The authors give detailed account of the method and its limitations. In particular they are able to show how a smooth phase-space distributions emerge from coarse graining.

My main question is whether the confinement is the main source of the integrability breaking in this system? Is it possible that other sources of non-integrability, like three body interactions due to transitions to higher transverse states, will lead to significant deviations of the idealized picture and whether they can be accounted for by suitable modifications of Generalized Hydrodynamic equations? Such discussion can be very valuable for realistic experiments.

The manuscript is of clear interest to the theoretical community studying dynamics of quantum systems close to integrability and the presented method can be used for efficient simulations of future experiments with ultra cold gases."

Answer: the referee asks “whether confinement is the main source of the integrability breaking in this system”.

First we need a qualification. Confinement breaks integrability globally. But at the Euler scale (with potentials varying on very large length scales), integrability is conserved in every local fluid cell, and the integrability-breaking effect of confinement is accounted for solely by a modification of the Euler scale equations by a force term. These are still equations for all conserved densities of the original, homogeneous model, derived in [16]. In the exact limit of the Euler scale, Euler equations with force terms are exact and valid at all times, and do not lead to thermalisation or the loss of information about the higher conserved quantities.

The fact that we still have as many equations as there are homogeneous conserved charges is not surprising; it is the same phenomenon as what happens in any usual Galilean-invariant fluid - such as water - in an external inhomogeneous potential - such as a gravitational field. Total momentum is broken by gravity, but momentum is still locally conserved, and the local momentum-conservation equation is still written, modified by a force term that breaks total momentum (the standard Euler equations with force term). See the explanations in [16], p 9.

Second, beyond the Euler scale, the potential does indeed break integrability locally and should lead to thermalisation. We have made an estimate of this rate on the basis of our simulations. We can do so by exploiting the notion of a “quantum jump” as explained in Appendix VIII A. For other sources of integrability breaking mentioned by the referee (three-body interaction, which may lead to losses, transitions to other transverse states, perhaps also inter-tube interactions in experiments where quasi-one-dimensionality is achieved by electromagnetic lattices of tubes), we have introduced the notion of a collision term in the GHD equations (see the penultimate section of the main text). For the particular case of inter-tube interactions, we sketch an estimate of the size of the collision term in Appendix VIII B.

---

## Round 3 · List of Changes

-we added a paragraph in the main text in page 6 where we briefly discuss mechanisms of integrability breaking.
-this is complemented by a more extended discussion in a new appendix VIII ("Timescales for integrability breaking")

You are currently on this page

Resubmission 1711.00873v3 on 30 May 2019

---

## Editorial Decision

published